# The Influence of Mitochondrial-DNA-Driven Inflammation Pathways on Macrophage Polarization: A New Perspective for Targeted Immunometabolic Therapy in Cerebral Ischemia-Reperfusion Injury

**DOI:** 10.3390/ijms23010135

**Published:** 2021-12-23

**Authors:** Sihang Yu, Jiaying Fu, Jian Wang, Yuanxin Zhao, Buhan Liu, Jiahang Wei, Xiaoyu Yan, Jing Su

**Affiliations:** Department of Pathophysiology, College of Basic Medical Sciences, Jilin University, Changchun 130021, China; yush19@mails.jlu.edu.cn (S.Y.); fujy21@mails.jlu.edu.cn (J.F.); wjian21@mails.jlu.edu.cn (J.W.); zhaoyx19@mails.jlu.edu.cn (Y.Z.); liubh20@mails.jlu.edu.cn (B.L.); weijh7019@jlu.edu.cn (J.W.)

**Keywords:** cerebral ischemia-reperfusion, mtDNA, inflammation, macrophages, immune metabolism, STING

## Abstract

Cerebral ischemia-reperfusion injury is related to inflammation driven by free mitochondrial DNA. At the same time, the pro-inflammatory activation of macrophages, that is, polarization in the M1 direction, aggravates the cycle of inflammatory damage. They promote each other and eventually transform macrophages/microglia into neurotoxic macrophages by improving macrophage glycolysis, transforming arginine metabolism, and controlling fatty acid synthesis. Therefore, we propose targeting the mtDNA-driven inflammatory response while controlling the metabolic state of macrophages in brain tissue to reduce the possibility of cerebral ischemia-reperfusion injury.

## 1. Introduction

Every year, a considerable number of patients worldwide die from acute brain dysfunction caused by cerebral ischemia-reperfusion injury [1,2]. After an ischemic stroke, the cells in the core infarct area suffer irreversible neuronal damage due to the energy supply falling below the threshold. The ischemic penumbra with perfusion range between the functional threshold and the low threshold for irreversible tissue morphological damage has the potential to restore tissue function [3,4,5]. However, the restoration of blood flow and oxygen supply during treatment usually aggravates tissue damage and causes severe inflammation, that is, ischemia-reperfusion injury [6,7].

The brain is extremely sensitive to energy supply, and mitochondria, which are the most abundant organelles in brain tissue, efficiently produce ATP to supply energy to the brain [8,9]. When ischemia-reperfusion occurs, the cells exhibit a wide range of mitochondrial dysfunction caused by apoptosis and related cascade reactions [10,11]. As a downstream effect of mitochondrial apoptosis, inflammation further destroys brain function. The opening of the mitochondrial permeability transition pore (mPTP) is deemed to be the core cause of this inflammatory disaster [6,11,12]. Cytochrome C is released to cause apoptosis. In addition, mitochondrial-related damage-associated molecular patterns (mtDAMPs), such as mtDNA, formyl peptide, ATP, and mtROS, are released outside the cell, which causes the innate immune response of the nervous system [13,14,15,16]. More and more inflammatory damage responses of mtDNA have been recorded. Free mtDNA activates toll-like receptors in membranes and endosomes, which promotes the activation and mutual activation of NOD-, LRR-, and pyrin-domain-containing protein 3 (NLRP3) inflammasome and cyclic GMP-AMP synthase (cGAS) stimulator of interferon genes (STING) [17,18,19]. Eventually, it expands the inflammatory effect in ischemia-reperfusion injury.

Mononuclear macrophages/microglia play an important role in the inflammatory response induced by cerebral ischemia-reperfusion. Under physiological conditions, they release neurotrophic factors and nerve growth factors and trim synapses [20]. When the ischemic stroke is reperfused, a large amount of mtDNA diffuses into the brain tissue around the infarct area, and the above-mentioned inflammatory pathways are activated in the macrophages, causing the innate immune response caused by neurotoxic macrophages [21,22]. However, with the in-depth development of metabolic immune research, the metabolic patterns of macrophages and their immune functions are thought to be complementary [23].

After reperfusion, the activation and infiltration of inflammatory cells, and the synthesis and secretion of adhesion molecules form a mutually promoting cascade reaction. In the primary and secondary injury stages of stroke, inflammation plays a major role. At this stage, the curative effect is mainly achieved by blocking the inflammatory cascade, including methods such as therapeutic hypothermia, hydrogen sulfide treatment, and the use of cyclosporine, cannabinoids, superoxide dismutase, metformin, and stem cell treatments [6].

Therefore, in the study of cerebral ischemia-reperfusion injury, the innate immune response of macrophages caused by mtDNA can be overlooked. Here, we summarized that the mitochondrial-related inflammatory response caused by mtDNA regulates the polarization direction of macrophages by changing the metabolic mode of macrophages, including glucose, lipid, and amino acid metabolism, and then regulates the process of ischemia-reperfusion injury. The possibility of reducing ischemia-reperfusion injury by controlling macrophage metabolism and mtDNA inflammation pathways as targets are proposed.

## 2. The Function of Macrophages in Cerebral Ischemia-Reperfusion

Necrosis induced by ischemic injury mainly occurs in brain endothelial cells and neurons, whereas apoptosis mainly occurs in neurons, with a definite hysteresis [24,25]. Damage-associated molecular pattern (DAMP) is released by necrotic cells, especially mtDAMP, which activates microglia/infiltrating macrophages [26]. Therefore, macrophage polarization behavior will extensively affect the degree of brain nerve damage and functional recovery [27]. Neuroprotective macrophages and neurotoxic macrophages in the ischemic penumbra have been classified into M1/M2 according to their cell function and metabolic mode, also known as the polarization model of classical activation/alternative pathway activation [28,29].

The activation of M1 macrophages by IFN-γ, LPS, and DAMPs involves pathways such as Janus kinase (JAK), signal transducer and activator of transcription 1 (STAT1), and NFκB. Toll-like receptors (TLRs), especially TLR4, respond to the above polarizing factors, and these processes increase the production of inflammatory mediators, such as TNFα, IL-23, IL-1β, IL-12, chemokines, NO, and ROS, In addition to its role in acute inflammation, M1 macrophages highly express major histocompatibility complex class Ⅱreceptor (MHC-Ⅱ) and CD86 to achieve antigen presentation function. However, stimulating factors of macrophages activated by alternative pathways are usually IL-4, IL-10, and IL-13. The signal transducer and activator of transcription 6 (STAT6), peroxisome proliferator-activated receptor γ(PPARγ), Jumonji-domain-containing 3(Jmjd3), and interferon regulatory factor 4(IRF4) pathways in the cell are also activated. High expression of TGF-β, CD163, fibronectin 1, and Arg1 promote tissue repair after inflammation [23]. However, microglia/macrophages are highly heterogeneous in the immune microenvironment of cerebral ischemic tissue, which usually involves the interaction between cells [30,31,32]. Therefore, cell polarization caused by a single stimulus does not exist, and the overall and exact macrophage phenotype should be elucidated. In addition to macrophages that have lost their basic functions due to ischemic injury, we categorize microglia/macrophages as neuroprotective or neurotoxic (Figure 1).

### 2.1. Neurotoxic Macrophages

After ischemia-reperfusion, the pro-inflammatory signal released by injured neurons initiates the microglia phagocytosis. UDP is an imperative “eat me” signal in addition to ATP, and it recognizes phagocytic neurons through the pyrimidinergic receptor, P2Y6 [33,34,35]. As a result, excessively activated phagocytic signals may excessively increase the clearance function of microglia and cause neuronal damage. Phagocytic microglia and macrophages express the phagocytic markers CD68 and NG2. The inflammatory phenotype of brain macrophages is often associated with NG2 and CD200 [36,37]. Compared with microglia, neurotoxic infiltrating macrophages (NG2−/CD200+ macrophages) release more mtROS, IL-1β, and NO. NG2+/CD200− macrophages and NG2−/CD200+ macrophages accumulate at the core of the pathological brain damaged by the BBB. CD200 expressed by neurons binds to macrophages expressing CD200 receptors, and at the same time inhibits the inflammatory response of macrophages; however, macrophages expressing CD200 (NG2−/CD200+ macrophages) release pro-inflammatory mediators, such as ROS and IL-1β, thus playing a harmful role [38,39].

### 2.2. Neuroprotective Macrophages

In the early stages of neuroinflammation, microglia/macrophages engulf apoptotic neuronal cells [40]. Kronenberg et al. sequenced blood-derived macrophages and microglia after cerebral ischemic injury in mice based on the classic M1 and M2 macrophage gene expression profiles, and the results showed that blood-derived macrophages are typically more inclined to the M2 neuroprotective phenotype [41]. However, Matsumoto et al. discovered brain Iba1+/NG2+ cells (BINCs, which appear as neuroprotective macrophages in the cerebral ischemic area). These macrophages have a strong proliferation capacity and secrete neuron growth factors [42]. Macrophages activated after injury are considered regionally heterogeneous, although the specific mechanism remains unclear. The mRNA expression of the phagocytic markers CD68 and F4/80 in the ventral horn was lower than that in the dorsal horn, indicating that the microglia in the dorsal horn had stronger phagocytic activity than the microglia in the ventral horn. Phagocytosis of the myelin sheath by dorsal horn microglia may be related to the occurrence of neuropathic pain. However, after a nerve injury, the activation of spinal microglia in the ventral and dorsal horns is accompanied by the upregulation of the pro-inflammatory cytokines IL1β and IL6. The expression of Arg1 in the dorsal horn increased, while the expression of CD206 and YM1 remained unchanged. Therefore, activated microglia induced by peripheral nerve injury cannot be clearly classified [43].

## 3. The Inflammatory Response Is Driven by MtDNA in Cerebral Ischemia-Reperfusion Injury

Mitochondrial injury is undoubtedly the central module that initiates cerebral ischemia-reperfusion injury(IR/I) [9], during which mtDAMPs are released [13], mtDNA drives the multipathway innate immune response and is an important marker for evaluating brain damage [44,45], The initiation of the innate immune response is triggered by the recognition of pathogen-associated molecular patterns (PAMPs) or DAMPs by the pattern-recognition receptor (PRR). In addition, endogenous cellular products are related to tissue damage and self-danger signals, such as heat shock proteins and defective nucleic acids [46,47]. PRR is located in the cell membrane and cytoplasm, and mainly includes the toll-like receptor family, the RNA helicase family, and the nucleotide binding and oligomerization-domain-like receptor (NLR) family. TLR9, NLRP3, and STING were considered in this study.

### 3.1. TLRs

TLRs are located on the cell surface or in endosomes [48]. The cytoplasmic PRR includes the RNA helicase family (retinoic-acid-inducing gene protein 1, RIG-1, and melanoma-differentiation-related gene 5, MDA5) and NLR family [49]. In TLR, TLR signals can be transmitted through MyD88-related or MyD88-independent signaling. In general, except for TLR3, the signals of all TLR family members are initiated by the MyD88 protein and are transmitted through MyD88-dependent signaling pathways [50]. In IR/I, damaged neurons release endogenous DAMPs to activate TLR, and further stimulate the inflammatory cascade to induce secondary damage [51]. MtDNA-TLR9 plays an important role in IR/I. Zhang et al. first discovered that mtDNA can be released into the blood and bind to TLR9 to mediate an immune response [14,52]. In subsequent studies, it was shown that increased myocardial mtDNA would activate the TLR9-dependent inflammatory response, leading to the death of mice after heart overload [53]. At the same time, there is evidence that TLR3 signaling is harmful to brain injury and middle cerebral artery occlusion (MCAO) 2 h after reperfusion for 22 h, the cerebral infarct area of TLR3- or TLR9-deficient mice did not significantly decrease, and the neurological score did not improve. It shows that TLR3 or TLR9 has no protective effect on MCAO-induced ischemic brain damage. Clinical studies have shown that the expression of TLR3 and TLR9 in the peripheral blood of patients with ischemic stroke has nothing to do with nerve damage, which further indicates that TLR3 and TLR9 are not directly involved in the regulation of cerebral ischemia-reperfusion inflammation damage [54,55,56]. Other TLRs are also widely involved in IR/I. The expression of TLR2 mRNA in focal ischemic brain tissue is significantly increased, and TLR2 was the most strongly upregulated TLR. In addition, compared with wild-type mice, the infarct size of TLR2-deficient mice was significantly reduced. In ischemic brain tissue, the common signaling pathway mediated by TLR2/1 dimer and CD36 induces ischemic inflammation and tissue damage. These results indicate that TLR2 plays an important role in inflammatory damage after cerebral ischemia [57,58,59]. Cao et al. found that infarct size and nerve damage of TLR4-knockout (KO) mice in focal MCAO were significantly reduced [60]. Subsequently, the global cerebral ischemia model further confirmed that the cerebral ischemic damage of TLR4-knockout mice was also significantly reduced [61], and in vitro experiments have shown that after oxygen–glucose deprivation (ODG), the survival rate of cortical neurons cultured in TLR4 KO mice was significantly improved [62]. TLR8 activation aggravates ischemic brain damage. In addition, clinical studies have shown that the expression of TLR8 in the peripheral blood of patients with ischemic stroke is significantly related to the prognosis of patients, which proves the close connection between TLR8 and cerebral ischemia-reperfusion injury [55,63].

### 3.2. NLRP3

Among the NLRs, NLRP3 is the most widely described [64]. Studies have shown that NLRP3 deficiency can improve the neurovascular damage in experimental ischemic stroke [65]. Structurally, NLRP3 is linked to caspase-1 through an apoptosis-related dot-like protein, which contains a caspase recruitment domain (ASC) to form a macromolecular complex, the NLRP3 inflammasome, in response to PAMPs or DAMPs, and induces the release of pro-inflammatory cytokines and caspase-1-dependent pyrolysis [66]. It is generally believed that the activation of the NLRP3 inflammasomes requires two steps: first, NFκB-dependent expression of IL-1β precursor and NLRP3 is induced to respond to transcriptional regulatory pattern recognition receptors (such as TLR) or pro-inflammatory cytokine receptors, which then trigger the assembly and activation of NLRP3 inflammasomes [67,68,69,70]. There are four main NLRP3 activation models: ROS generation, lysosomal damage, cytoplasmic potassium efflux, and oxidized mtDNA released during programmed cell death, which is discussed in this article [71,72,73,74]. The interaction between mtDNA released by mitochondria and cardiolipin is also involved in the assembly and activation of NLRP3 inflammasomes [75]. With a few exceptions, K+ efflux has been identified as a common feature of NLRP3 activation [76]. After NLRP3 is activated, its oligomerization leads to the aggregation of its N-terminal pyran domain and the recruitment of the caspase recruitment domain (ASC). ASC interacts with procaspase-1, resulting in the formation of active caspase-1. This induces the secretion of active pro-inflammatory cytokines such as IL-1β and IL-18 [77]. Savage and colleagues found that after middle cerebral artery occlusion in mice caused cerebral ischemia, the production of IL-1 promoted the increase of IL-6 and CXCL1 levels. DAMPs enhance brain inflammation by directly stimulating the production of glial-cell-derived inflammatory mediators [78]. BTK is a tyrosine kinase that participates in the activation of the NLRP3 inflammasome, which in turn leads to the activation of caspase-1 and the production of mature IL-1β in the process of cerebral ischemia. Ito and colleagues showed that ibrutinib, a potent BTK inhibitor, inhibits NLRP3 inflammasome signaling in a focal brain ischemia-reperfusion model. Specifically, ibrutinib exerts a neuroprotective effect by inhibiting the maturation of IL-1β by suppressing the activation of caspase 1 in infiltrating macrophages and neutrophils in the infarct area [79].

### 3.3. STING

Studies have shown that the cGAS-STING pathway is involved in neuroinflammation and brain damage induced by I/R [80]. cGAS detects DNA in the cytoplasm to avoid the continuous activation of self-DNA in the nucleus. cGAS is believed to be mainly present in the cytoplasm, and studies have also shown that it exists in the nucleus and plasma membrane [81,82,83]. It combines with double-stranded DNA to form dimers [84,85]. cGAS then undergoes a conformational change, which helps convert ATP and GTP into 2′3 cyclic GMP-AMP (cGAMP) [86]. cGAMP is a second messenger, which binds to the stimulator of the endoplasmic reticulum resident protein of the interferon gene (STING) to induce conformational changes in its C-terminal tail. TANK-binding kinase 1 (TBK1) is recruited into STING to phosphorylate it and the transcription factor interferon regulatory factor 3 (IRF3), triggering the transcription of hundreds of interferon-stimulated genes (ISG) [87]. Type I IFN alpha receptor 1 (IFNAR1) or IRF3KO have neuroprotective effects on brain injury induced by transient middle cerebral artery occlusion (tMCAO) in mice [88,89]. Liao et al. found that HDAC3 transcriptionally promotes the expression of cGAS and enhances the activation of the cGAS–STING pathway by regulating the acetylation and nuclear localization of p65 in microglia. In vivo studies have shown that the absence of cGAS or histone deacetylase 3 (HDAC3) in microglia can reduce neuroinflammation and brain damage induced by ischemia-reperfusion [80]. Studies also have shown that STING is an important signaling molecule in immunity and inflammation and an important regulator of autophagy [90]. Therefore, STING-mediated autophagy may be involved in ischemia-reperfusion injury. At the same time, the cGAS–STING axis detects DNA in the cytosol, induces cell death programs, and initiates potassium efflux upstream of NLRP3, thereby participating in the activation of NLRP3 inflammasomes [91].

Ischemia-reperfusion causes cell pyrolysis, apoptosis, inflammatory storm, and diffusion of inflammatory substances after reperfusion, which is one of the causes of secondary neuron damage. As mentioned, mtDNA can bind to TLR9 and directly participate in cerebral ischemic injury, and it can further activate NLRP3 to induce inflammation and cause pyrolysis. The STING pathway also plays an important role in the activation of NLRP3, which can promote the outflow of potassium ions and lead to the activation of the former. MtDNA can also directly act on NLRP3 and STING (Figure 2). In short, many pathways play a role in expanding the inflammatory response in ischemia-reperfusion, and the close coordination of TLRs, NLRP3, and STING plays an important role in the pathological process of ischemia-reperfusion injury.

## 4. The Influence of MtDNA-Mediated Inflammation on the Metabolic Pattern of Macrophages

Different polarization directions of macrophages are involved in the maintenance of inflammatory response and tissue homeostasis in the body. Among them, the M1/M2 model represents the two extremes of macrophages driving polarization. M1 macrophages are activated by LPS/IFN-γ, which is a classic activation pathway, and anaerobic glycolysis and pentose phosphate pathways, which are the main metabolic pathways. M2 macrophages are activated by IL-4/IL-13, which is known as the “alternative activation pathway”. It has a complete tricarboxylic acid (TCA) cycle and oxidative phosphorylation (OXPHOS) process, with prominent fatty acid synthesis and metabolism. Although the molecular mechanism of macrophage polarization has not been elucidated in vivo, previous studies have distinguished M1/M2 through arginine metabolism, specifically involving the expression of two enzyme genes with arginine as a common substrate. That is, inducible nitric oxide synthase (iNOS) with high expression of M1 and arginase 1 (ARG1) with high expression of M2 [92,93] (Figure 1).

### 4.1. cGAS/STING

Studies on the regulation of macrophage polarization are often discussed in infection models by cGAS–STING. In Brucellosis, STING changes the macrophage metabolism mode through HIF1-α. The specific mechanism involves an increase in the succinic acid content, which increases the production of mitochondrial ROS, stabilizes the conformation of HIF1-α, and increases the transcription of related metabolic genes. The specific manifestation is that the glycolytic function of HIF is knocked out and the expression of anti-inflammatory genes increases [94]. Another interesting study has found that cGAS–STING affects the macrophage arginine metabolism. The host protective response including ornithine binding protein and inflammasome activation caused by *Brucella* does not depend on cGAS–STING. After the cGAS–STING pathway was inhibited, the expression of iNOS decreased and the expression of arginase increased. Lipid metabolism and its products play a critical role in regulating inflammation and regressing macrophage function [95] (Figure 3). The 4-hydroxynonenal (4-HNE) levels are significantly reduced in STING^−/−^ intestinal tissues. Differential regulation of genes and enzymes related to arachidonic acid metabolism has been noted in polarized macrophages [96]. For example, M1 macrophages showed that significant induction of the microsomal isoforms of COX2 PGE synthase (mPGES), COX1, leukotriene A4 hydrolase, thromboxane A synthase 1, and arachidonic acid 5-lipoxygenation enzyme (ALOX5) are downregulated. This is related to the increased metabolism of arachidonic acid and the production of eicosanoids in M1 macrophages. In fact, the upregulation of COX2 and cholesterol precursor sterol is functionally related to the expression of inflammatory genes in these cells. In contrast, IL4-activated M2 macrophages showed arachidonic acid 15-lipoxygenase and COX1, but mPGES was downregulated [96,97]. Therefore, activating different lipid metabolisms in macrophages can help these cells transition from an inflammatory phenotype to a tissue repair phenotype. Finally, lipid metabolism and fatty acids can promote and regulate the phagocytosis of macrophages. Lipidomic analysis of activated macrophages should further clarify the differential regulation of various lipid mediators, their metabolism, and the relationship between these and macrophage functions. However, the regulatory relationship between STING and lipid metabolism has not yet been explored. Interestingly, when cGAS expression was reduced, the metabolic patterns of macrophages in these studies were not significantly affected.

### 4.2. NLRP3 Inflammasome

The NLRP3 inflammasome is first activated in microglia shortly after brain ischemia-reperfusion injury, and then expressed primarily in neurons and in microvascular endothelial cells but mainly in neurons. Mitochondrial dysfunction plays an important role in the activation of microglial NLRP3 inflammasome after oxygen and glucose deprivation–reperfusion OGD/R [98]. In fact, citric acid and succinic acid, the by-products of macrophage activation and metabolic transfer, are the metabolic signaling factors and main molecules that regulate inflammatory signals [99]. The accumulation of succinic acid increased HIF-1α-induced NLRP3 inflammasome activation in a rat model of rheumatoid arthritis. A recent study showed that the NLRP3 inflammasome is an important regulator of glycolysis in M1 macrophages, and it increases PFKFB3 in an IL-1β-dependent manner [100]. Palmitate is one of the most abundant pro-inflammatory factors in plasma after the consumption of a high-fat diet. It can induce the activation of NLRP3 inflammasomes in bone-marrow-derived macrophages in mice, indicating that pro-inflammatory factors are very effective in activating NLRP3 inflammasomes. This highlights the fundamental mechanism between the elevation of pro-inflammatory factors and the subsequent progression of metabolic diseases [99].

### 4.3. TLRs

TLR4 participates in the inflammatory polarization of macrophages, and the activated downstream pathways involve NFκB, STAT, and MAPK, which leads to a significant increase in aerobic glycolysis and interruption of the TCA cycle to drive a specific pro-inflammatory response [101]. A key component that activates the metabolic transformation of macrophages is the glycolytic enzyme pyruvate kinase M (PKM), which exists as two alternatively spliced isoforms (PKM1 and PKM2). PKM1 catalyzes the last step of glycolysis by converting phosphoenolpyruvate into pyruvate, while PKM2 also promotes the expression of genes involved in glycolysis and inflammation. However, the molecular composition required for PKM2-dependent inflammatory functions is not well understood. Gupta et al. found that class IIa HDACs, especially HDAC7, regulate the function of the glycolytic enzyme PKM2, providing a molecular link between TLR-induced aerobic glycolysis and macrophage inflammation [102]. In addition, TLR activation can also induce the synthesis of non-classical eicosanoids, such as lipoxins that support the anti-inflammatory activity and pro-degradation function of macrophages [103]. At the same time, studies on TLR directly regulating the function of macrophages have shown that TLR4 activation inhibits autophagy by inhibiting FOXO3 and impairs the phagocytic ability of microglial [104].

Of course, it is not only the polarization state of macrophages in the ischemic area that affects the repair of brain function, but the state of macrophages themselves also affects cerebral ischemia-reperfusion injury. When macrophage stimulating 1 (MST1) is knocked out in microglia, it will alleviate mouse cerebral ischemia-reperfusion injury [105]. At the same time, it does not mean that the mtDNA-driven inflammation pathway can extensively and uniquely change the polarization state of microglia (Figure 2). In the study of spinal cord nerve injury, the inhibition of the HMGB1/NFκB axis can inhibit the pro-inflammatory polarization of microglia, which has a neuroprotective effect [106]. However, when AMPK/Nrf2 is activated synergistically, it can increase the polarization of macrophages in the M2 direction to reduce brain inflammation after stroke [107].

**Figure 2 ijms-23-00135-f002:**
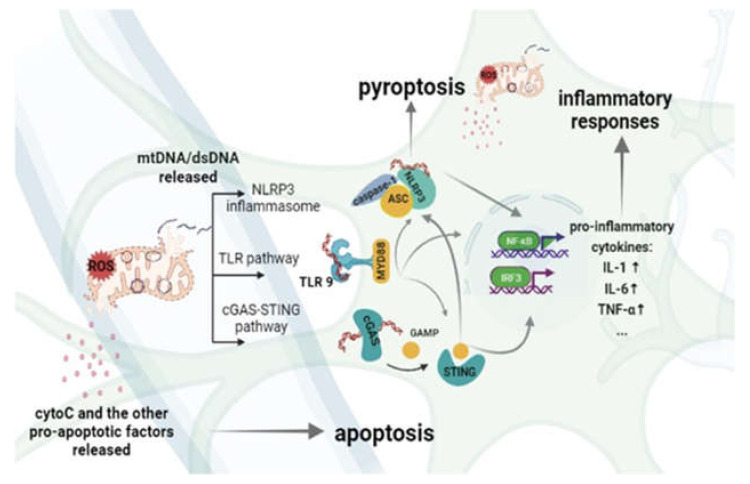
The main inflammatory response is mediated by mtDNA/dsDNA in IR/I. When mitochondria are damaged by oxidative stress, the permeability of the mitochondrial membrane will increase, and the leakage of cytochrome c and other apoptosis-inducing molecules in the mitochondrial matrix will initiate the apoptosis pathway. Then, the released mtDNA/dsDNA causes a series of inflammatory reactions: NLRP3 recruits caspase-1 (caspase-1) to form inflammasomes through the caspase recruitment domain (ASC), and it induces cell burns. On the one hand, it responds to the pro-inflammatory signal of TLR; TLR can directly recognize mtDNA/dsDNA through the MYD88 pathway, cGAS recognizes mtDNA in the cytoplasm, produces cGAMP, promotes the dimerization and activation of STING, and STING can also promote the outflow of K+ ions to participate in the NLRP3 pathway. The resulting inflammatory signal can also further activate the TLR located in the endosome, so that inflammation can be magnified. Finally, the NFκB and IRF3 pathways induce the production of inflammatory signals, leading to the massive release of cytokines IL-1, IL-6, TNF-α, etc., and triggering an inflammatory response.

## 5. The Therapeutic Prospects of Cerebral Ischemia-Reperfusion Targeting MtDNA-Mediated Inflammation and Microglia/Macrophage Metabolism

Excessive activation of microglia/macrophages can aggravate the inflammatory response of the central nervous system and cause damage after stroke reperfusion. The specific manifestation is that many patients receive treatment with different degrees of sequelae of the nervous system [108]. Therefore, laboratories have attempted to control the intensity of the inflammatory response and change the polarization of microglia/macrophages [109], although these potentially effective drugs have not yet been applied clinically. Here, we summarize the research progress of the main signaling pathway in the intracellular innate immunity, that is, the relevant inhibitors of the mtDNA inflammatory pathway and the application prospects of drugs targeting the immune metabolism of macrophages (Table 1).

### 5.1. Therapies That Target TLRs

The activation degree of TLR4 and downstream pathways such as NFκB is often used as a criterion for detecting the inflammatory environment of ischemia-reperfusion. In some animal models, creatine can effectively reduce ischemia-reperfusion injury in the heart, brain, and lungs [110,111,112]. Garcinol reduces cerebral ischemia-reperfusion injury by downregulating TLR4-NFκB [113]. Similarly, Kudiezi injection improves the cerebral ischemia-reperfusion injury of rats by downregulating the TLR4-NFκB pathway [114]. Pregabalin, as a drug for the treatment of neuropathic pain, has also been applied to the rat reperfusion model of hyperglycemia stroke model [115]. As a common anesthetic, propofol can also inhibit the above pathways and exert a neuroprotective effect [116]. Lactoferrin can inhibit TLR4-related pathways triggered by hypoxia–reoxygenation and ischemia-reperfusion, suggesting that lactoferrin can be used as a dietary intervention for cerebral ischemia [117].

Glucagon-like peptide-1 (GLP-1) is an intestinal peptide with a variety of physiological effects. One of its main functions is to regulate blood sugar levels when the blood sugar rises, since it promotes the release of insulin. GLP-1 can protect neurons from damage caused by neurodegenerative diseases. Lixisenatide is a GLP-1 analog and has a strong affinity with the GLP-1 receptor [118]. Experimental animal studies have shown that it can protect hippocampus CA1 neurons after cerebral ischemia by downregulating TLR2/4–NFκB signaling and upregulating p38/ERK pathway [119]. Some traditional Chinese medicine extracts have also been shown to have neuroprotective effects in stroke models, such as tangeretin, QiShenYiQi, and nobiletin [120,121,122]. In the study of Li. et al., increasing the level of microRNA-129-5p can improve neuroinflammation and blood–spinal cord barrier injury after ischemia-reperfusion by inhibiting the HMGB1 and TLR3 pathways [123]. Argon can downregulate the downstream transcription factors NFκB and STAT3 of TLR to exert neuroprotection. Thereby changing the mechanism of apoptosis (such as the generation of mitochondrial membrane potential and ROS) and realizing the neuroprotection of reperfusion after ischemia [124]. Andrographolide derivative CX-10 has a neuroprotective effect on transient focal ischemia in rats through the Nrf2/ARE and TLR/NFκB signaling pathways [125].

### 5.2. Therapies That Target NLRP3

The NLRP3 inflammasome is thought to mediate the inflammatory response during ischemia-reperfusion injury. Mitochondrial dysfunction plays an important role in the activation of the NLRP3 inflammasome pathway in primary microglia and BV2 cells after OGD/R. Diazoxide as a mitochondrial protective agent and sulforaphane can effectively alleviate the response of NLRP3 inflammasome in primary microglia and BV2 cells [98,126]. Thrombolytic therapy is still challenging in patients with hyperglycemia, because it is associated with poor prognosis and increased bleeding turnover. Tissue plasminogen activator (tPA)–induced cerebrovascular damage is related to the upregulation of thioredoxin-interacting protein (TXNIP), and upregulation of thioredoxin-interacting protein (TXNIP) has harmful effects on hyperglycemia. Verapamil strongly reversed the tPA-induced activation of TXNIP/NLRP3 inflammasomes and reduced infarct size [127]. Chemokine-like factor 1, CKLF1, is a potential target for the treatment of ischemic stroke. The compound IMM-H004, a new coumarin derivative, downregulated the amount of CKLF1 binding with C–C chemokine receptor type 4, further suppressing the activation of the NLRP3 inflammasome and the following inflammatory response, thereby improving brain ischemia-reperfusion injury [128]. NLRP3 inflammasome-mediated pyrolysis plays a key role in the pathogenesis of cerebral ischemia-reperfusion injury. Therefore, some studies usually reduce brain damage by reducing the pyrolysis caused by NLRP3, e.g., hispidulin, astilbin, and adiponectin peptides inhibit NLRP3-mediated pyrolysis by regulating the AMPK/GSK3β signaling pathway, improve the neurological symptoms after ischemia-reperfusion injury in rats, and reduce the infarct size and cerebral edema [129,130,131]. Of course, using chemical inhibitors SB216763 and GSK-3β siRNA to inhibit the activation and expression of GSK-3β in vivo has similar neuroprotective effects [132]. However, spautin-1 is an inhibitor of USP10 and USP13, an inhibitor of autophagy. Treatment with Spautin-1 can reduce autophagy and ROS accumulation and alleviate NLRP3-inflammasome-dependent pyrolysis [133], although the molecular mechanism between autophagy and pyrolysis is still unclear. Another interesting study is that exosomes derived from bone marrow mesenchymal stem cells can inhibit NLRP3-mediated inflammation and pyrolysis by regulating the polarization of microglia, which can improve brain I/R injury [134]. Although the mechanism is not clear, l-homocarnosine reduces the inflammatory response in cerebral ischemia-reperfusion injury by reducing the expression of NLRP3 [135]. Kv1.3 channel blocker, PAP-1, reduces cerebral ischemia-reperfusion injury by remodeling the M1/M2 phenotype and inhibiting the activation of microglia NLRP3 inflammasome. However the connection between NLRP3 and macrophage polarization was not explained [136]. TMEM59 (also known as dendritic-cell-derived factor 1, DCF1) is a type I transmembrane protein. Increasing the expression of TMEM59 effectively prevents the pyrolysis and inflammation of microglia after OGD/R treatment [137].

For cerebral ischemia-reperfusion therapy targeting NLRP3, its small molecule inhibitor, MCC950 [138,139], has been proposed. When MCC950 is co-treated with low-temperature nerve cells, it activates PTEN and the PI3K/Akt/GSK-3β signaling pathway, which then reduced nerve cell pyrolysis [140]. This drug has also been used in the treatment of a diabetic mice cerebral ischemia-reperfusion model [141]. Chlorpromazine and promethazine, idebenone, and resveratrol confer neuroprotective effects on stroke by inhibiting the activation of NLRP3 inflammasomes [142,143,144]. In a mouse model of global cerebral ischemia, phthalide derivative CD21 significantly inhibits the activation of HMGB1-mediated TLR4/NFκB signaling pathway and cathepsin B-mediated NLRP3/ASC/Caspase-1 signaling pathway, thereby inhibiting the NLRP3 inflammasome [145]. However, meisoindigo (3,30-linked bisindole, a second-generation derivative of indirubin) may inhibit the activation of NLRP3 inflammasomes and M1 polarization by inhibiting the TLR/NFκB signaling pathway, and it is expected to become a new drug for the treatment of ischemic stroke [146]. The bromodomain-containing protein 4 (BRD4) selective inhibitor JQ1 can reduce the infarct volume, brain water content, and neurological deficit score of MCAO mice. BRD4 is a member of the bromo and extra-terminal (BET) family, which promotes inflammation in various tissues and cells. The expression of BRD4 is related to glial cell activation and brain ischemia-reperfusion injury in mice with MCAO [147].

Some plant-derived drugs have also been widely used in models of cerebral ischemia-reperfusion injury. Anthocyanins, tomentosin, and procyanidins in bayberry can simultaneously downregulate the inflammatory pathway mediated by TLR and NLRP3 [148,149,150]. Salvianolic acids may play a neuroprotective effect by reducing neuronal apoptosis, changing the phenotype of microglia from M1 to M2, and inhibiting the NLRP3 inflammasome/pyrolysis axis of microglia [151]. Glycosides in the Buyang Huanwu decoction decrease the death rate after cerebral ischemia-reperfusion injury in rats [152]. Menopausal women have a higher incidence of stroke than men of the same age, and estrogen is believed to be the main cause of this difference. However, the result of estrogen replacement therapy in preventing postmenopausal stroke is controversial, and it has caused widespread controversy due to its serious side effects after long-term use. Genistein (Gen) is a natural phytoestrogen with minor side effects and has a protective effect against cerebral ischemia injury [121].

### 5.3. Therapies That Target cGAS/STING

Studies on ischemia-reperfusion in the liver, brain, intestines, and other organs have proved that the STING−/− group has a lower degree of damage [96,153,154], although gene editing methods have almost lost their applicability in clinical practice, it is undeniable that STING as a target for the treatment of ischemia-reperfusion injury is of great research significance. Decout et al. summarized in detail the small-molecules inhibitors targeting the CGAS/STING pathway, including cGAS inhibitor(s), dsDNA binding inhibitors, and STING inhibitors [155]. Lin et al. proposed that 25-hydroxycholesterol (25-HC) protects against cerebral ischemia-reperfusion injury by inhibiting STING activity. The specific mechanism is that 25-HC promotes autophagy through mTOR to increase the degradation of STING autophagy pathway. This reduces the neuroinflammatory response. Therefore, increasing the degradation of STING from the autophagy or proteasome pathway can effectively reduce the damage after ischemia and revascularization [156]. Shen et al. believe that microRNA-24-3p reduces liver ischemia-reperfusion injury in mice by inhibiting STING signaling [157]. Liao et al. downregulated the transcription of cGAS by NFκB by inhibiting HDAC3 to weaken the inflammatory direction polarization of microglia in the cGAS/STING pathway [80]. Jie Wu et al. proposed that liproxstatin-1 can downregulate cGAS/STING by reducing lipid peroxidation to reduce intestinal ischemia-reperfusion injury [154], although the mechanism of lipid peroxidation and STING activation has not been explored. An oligonucleotide, A151, was found in the study of Li et al. that can downregulate the expression of cGAS to reduce the inflammatory response of microglia [158]. Research on these inhibitors related to the source, path, and activity of cGAS/STING is still ongoing.

### 5.4. Therapies That Target Microglia/Macrophage Metabolism

Immunity and metabolism are two integrated systems. As discussed, pro-inflammatory microglia/macrophages can cause neuronal damage through the release of inflammatory cytokines, ROS, and excessive phagocytosis. This inflammatory microglia/macrophage has typical metabolic pattern characteristics, such as high-level glycolysis, TCA cycle block, or NOS-led arginine metabolism [159]. Therefore, by inhibiting these metabolic pathways, the innate immune functions can be arrested to reduce nerve damage.

Some glycolysis inhibitors such as STF31 (GLUT1 inhibitor), 2-DG, and HK1 inhibitors can reduce the inflammatory damage caused by microglia [160,161,162]. Astragaloside IV monomer extracted from traditional Chinese medicine was used in animal experiments. Astragaloside IV promotes microglia/macrophage M2 polarization and then enhances neurogenesis and angiogenesis through the PPARγ pathway after cerebral ischemia-reperfusion injury in rats [163]. Another drug, 3-BU, is a benzodiazepine sedative and hypnotic drug. It reduces the level of cell metabolism by simultaneously inhibiting glycolysis and oxidative phosphorylation of microglia/macrophages. It has a good therapeutic effect in the acute phase of rat stroke model [164]. Some studies believe that mitochondrial autophagy will promote the glycolysis process, and mitochondrial autophagy inhibitors such as CsA can effectively inhibit the transformation of the M1 phenotype of macrophages [11,12]. L-Monomethylarginine (L-NMMA) is an inhibitor of nitric oxide synthase. By reducing the release of NO and peroxynitrate from pro-inflammatory macrophages, it reduces the symptoms of myocardial ischemia treatment [165]. Mitochondrial dysfunction is a common feature of the pathophysiology of central nervous system injury. Mitochondria usually undergo repeated division and fusion to maintain metabolic homeostasis and cell health [166]. Under normal circumstances, the balance of division and fusion is important for the reorganization of mitochondrial components, the removal of damaged substances, and the maintenance of mitochondrial health. Mitochondrial dynamics imbalance can disrupt mitochondrial function. DRP1 activated by the TLR4 signal increases the mitochondrial division of microglia and induces metabolic reprogramming from OXPHOS to glycolysis, leading to pro-inflammatory activation [167]. As a result, the mitochondrial division/DRP1 inhibitor (Midivi-1) was used in a rodent brain inflammation treatment model [168]. Abe et al. proposed that the combination of controlling microglia/macrophage metabolism in the hyperacute phase of brain trauma or stroke and enhancing neuroprotection in the acute to subacute phase may become a feasible treatment strategy [28].

## 6. Conclusions and Perspectives

Free mtDNA stimulates multiple pathways of inflammation including TLR9, NLRP3, and STING, leading to apoptosis, necrosis, and the inflammatory cycle of the immune microenvironment [25]. As such, it plays an important role in severe trauma, non-hemolytic transfusion reactions, and ischemia-reperfusion injury discussed in this article [169]. However, some problems still exist, such as the heterogeneity of the polarization level of macrophages in the injured area and the difficulty of defining the administration time in the acute and subacute phases in the experiment [170]. However, the polarization state of macrophages/microglia in the immune microenvironment of the brain does affect the prognostic effect after reperfusion. Although many experimental studies on ischemia-reperfusion injury have been carried out around the world, more molecular mechanisms related to injury have not been explored in depth; and there is a lack of effective drugs and therapies for precise treatment of it. Moreover, the blood–brain barrier will reduce the efficacy of drugs to a certain extent.

There is no doubt that the innate immune response signal driven by mtDNA polarizes macrophages in the pro-inflammatory direction, but the molecular mechanism is rarely summarized or explored. Therefore, we used metabolic immunity as a framework to summarize the changes in mtDNA-driven inflammatory signals on the metabolic pattern of macrophages, with the aim of stabilizing or reversing the metabolic pattern of macrophages in clinical treatment and basic research to control the inflammatory injury after cerebral ischemia-reperfusion.

**Figure 3 ijms-23-00135-f003:**
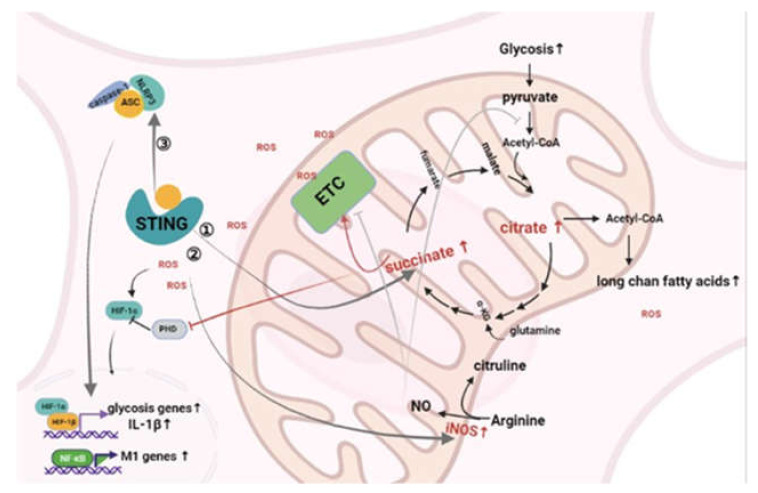
STING-centered inflammation pathway changes the metabolism of macrophages. ① Activated STING increases the content of succinic acid, promotes ETC to produce more ROS, stabilizes the structure of HIF-1, and ultimately promotes the expression of glycolytic genes. ② Activated STING increases the expression of iNOS and promotes the production of NO from arginine. ③ cGAS/STING can promote the activation of NLRP3 inflammasomes through K+ efflux, and promote the transcription of glycolytic genes mediated by IL-1β.

**Table 1 ijms-23-00135-t001:** mtDNA driven inflammation and macrophage metabolism targeted therapy for cerebral ischemia-reperfusion injury. ↑: Activation/Upregulation; ↓: Inhibition/Downregulation.

Target	Inhibitor	Mechanism	Models	Reference
TLRs	Garcinol	TLR4-NFκB↓	Middle cerebral artery occlusion/reperfusion (MCAO/R), oxygen–glucose deprivation and reperfusion (OGD/R)	[113]
	Kudiezi injection	TLR4-NFκB↓	Rat models of transient middle cerebral artery occlusion (tMCAO)	[114]
	Pregabalin	HMGB1/TLR4-NFκB↓	Middle cerebral artery occlusion (MCAO) model	[115]
	Propofol	TLR4-NFκB↓	Retinal ischemia reperfusion injury (RIRI)	[116]
	Lactoferrin	TLR4-related pathways↓	Anoxia and reoxygenation cell model, Institute for Cancer Research (ICR) mice	[117]
	Lixisenatide	TLR4-NFκB↓ P38/ERK↑	Not applicable	[118]
	Tangeretin	Inflammatory cytokine brain injury markers↓	Mice model of cerebral ischemia/reperfusion injury	[120]
	QiShenYiQi	TLR4-NFκB↓	OGD/R	[121]
	Nobiletin	Akt/mTOR↑ TLR4–NFκB↓	MCAO	[122]
	Argon	TLR2/TLR4/STAT3/NFκB↓	Retinal ischemia reperfusion injury (IRI) in rats	[124]
	CX-10	Nrf2/AE TLR/NFκB	Rat models of middle cerebral artery occlusion/reperfusion (MCAO/R)	[125]
NLRP3	Diazoxide	NLRP3 inflammasome activation↓	Transient middle cerebral artery occlusion (tMCAO) rat model, oxygen–glucose deprivation/reoxygenation (OGD/R)	[98]
	Sulforaphane	NLRP3 inflammasome activation↓	Middle cerebral artery occlusion (MCAO) model	[126]
	Verapamil	NLRP3 inflammasome activation↓	Transient middle cerebral artery occlusion (MCAO)	[127]
	IMM-H004	NLRP3 inflammasome activation↓	PMCAO model of focal ischemia	[128]
	Hispidulin	AMPK/GSK3β/NLRP3- pyrolysis↓	Middle cerebral artery occlusion (MCAO), oxygen–glucose deprivation/reoxygenation (OGD/R)	[129]
	Astilbin	AMPK/GSK3β/NLRP3- pyrolysis↓	middle cerebral artery occlusion (tMCAO) model with C57BL/6 J mice, oxygen–glucose deprivation and reintroduction (OGD-R) model	[131]
	Adiponectin peptide	AMPK/GSK3β/NLRP3- pyrolysis↓	middle cerebral artery occlusion-reperfusion (MCAO/R) model in rats	[132]
	SB216763	AMPK/GSK3β/NLRP3- pyrolysis↓	Middle cerebral artery occlusion–reperfusion (MCAO/R) model in rats, oxygen–glucose deprivation/reoxygenation (OGD/R)	[133]
	Spautin-1	Autophagy/NLRP3- pyrolysis↓	Middle cerebral artery occlusion–reperfusion (MCAO/R) model in rats, oxygen–glucose deprivation/reoxygenation (OGD/R)	[134]
	l-Homocarnosine	NLRP3 expression↓	Middle cerebral artery occlusion/reperfusion (MCAO/R) model in rats	[135]
	PAP-1	M1 polarization↓ NLRP3 inflammasome activation↓	Middle cerebral artery occlusion/reperfusion (MCAO/R) model in rats and oxygen–glucose deprivation/reoxygenation (OGD/R) in primary microglia	[136]
	TMEM59	NLRP3- pyrolysis↓ NLRP3-inflammation↓	middle cerebral artery occlusion (MCAO), oxygen–glucose deprivation/reperfusion (OGD/R)	[137]
	MCC950	NLRP3- pyrolysis↓	Oxygen–glucose deprivation/reoxygenation (OGD/R)	[140]
	Chlorpromazine and promethazine	NLRP3 inflammasome activation↓	Middle cerebral artery occlusion/reperfusion (MCAO/R) model in rats	[142]
	Idebenone	NLRP3 inflammasome activation↓	Oxygen–glucose deprivation and reperfusion (OGD/R)	[143]
	Resveratrol	NLRP3 inflammasome activation↓	Middle cerebral artery occlusion/reperfusion (MCAO/R) model in rats	[144]
	CD21	HMGB1/TLR4-NFκB↓ Cathepsin B/NLRP3 inflammasome activation↓	Global cerebral ischemia–reperfusion model in mice	[145]
	Meisoindigo	TLR4-NFκB↓ Cathepsin B/NLRP3 inflammasome Activation↓ M1 polarization↓	C57BL/6J mice middle cerebral artery occlusion (MCAO) stroke model, HT-22 and BV2 cells in vitro oxygen–glucose deprivation/reperfusion (OGD/R) model	[146]
	JQ1	NLRP3- pyrolysis↓ NLRP3- inflammation↓	Middle cerebral artery occlusion (MCAO) model	[147]
	Tomentosin	TLR and NLRP3 inflammation-related pathways↓	Rats Cerebral ischemia–reperfusion model, oxygen–glucose deprivation/reperfusion (OGD-R)	[148]
	Anthocyanin	TLR and NLRP3 inflammation-related pathways↓	ICR mice cerebral ischemic/reperfusion (I/R) injury	[149]
	Procyanidins	TLR and NLRP3 inflammation-related pathways↓	Middle cerebral artery occlusion/reperfusion (MCAO/R), oxygen–glucose deprivation/reoxygenation BV2 cells	[150]
	Salvianolic acids	M2 polarization↑ Neuronal apoptosis ↓ NLRP3- pyrolysis↓	Middle cerebral artery occlusion/reperfusion (MCAO/R) model in rats, oxygen–glucose deprivation/reoxygenation (OGD/R) model	[151]
	Buyang Huanwu decoction	NLRP3- pyrolysis↓	Rats focal cerebral ischemia and reperfusion model	[152]
	Genistein (Gen)	NLRP3 inflammasome activation↓	Not applicable	[153]
cGAS/STING	25-HC	Autophagy↑ STING degradation↑	Liver I/R model	[157]
	MicroRNA-24-3p	Inhibit cGAS/STING	Transient middle cerebral artery occlusion (tMCAO)	[158]
	Liproxstatin-1	Inhibit cGAS/STING	Not applicable	[155]
	A151	cGAS expression↓	Not applicable	[159]
Macrophage metabolism	STF31	Glycolysis↓	Diabetic retinopathy (DR) model	[161]
	2-DG	Glycolysis↓	not applicable	[162]
	HK1 inhibitor	Glycolysis↓	Middle cerebral artery occlusion/reperfusion (MCAO/R) model in rats	[163]
	Astragaloside IV	PPARγ- M2 polarization↑	Rat traumatic brain injury (TBI) model	[164]
	CsA	Mitophagy↓- Glycolysis↓ M2 polarization↑	Not applicable	[11,12]
	3-BU	Glycolysis↓ OXPHOS↓	Not applicable	[165]
	L-NMMA	Inhibit iNOS	Not applicable	[166]
	Midivi-1	Mitochondrial division↓ Glycolysis↓	Not applicable	[169]

## Author Contributions


S.Y.: Draft and figures preparations; J.W. (Jian Wang) and J.F.: literature searching; Y.Z., B.L. and J.W. (Jiahang Wei): review and editing; J.S.: review, study design and funding acquisition; X.Y.: conceptualization, project administration, funding acquisition. All authors have read and agreed to the published version of the manuscript.

## Figures and Tables

**Figure 1 ijms-23-00135-f001:**
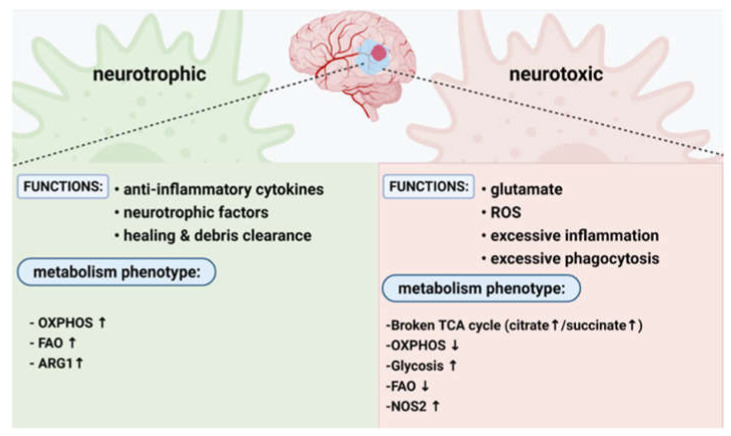
The functional and metabolic phenotype of ischemic core microglia/macrophages. Neuroprotective macrophages: (1) secretion anti-inflammatory factors, (2) secretion of neuroprotective/growth factors, and (3) tissue repair and tissue debris removal. Oxidative phosphorylation increases, fatty acid oxidation increases, and arginine metabolism. Neurotoxic macrophages: (1) neuronal damage caused by the excitotoxicity of glutamate released by neurons and microglia; (2) reactive oxygen species (ROS) on proteins, phospholipids, and neurons; nucleic acid is chemically modified; (3) neuronal damage caused by excessive inflammation related to local and systemic immune responses; (4) neuronal cell death due to excessive phagocytosis of immune cells. The kinetic energy of the tricarboxylic acid cycle is insufficient (causing the accumulation of citric acid and succinic acid), therefore, the level of oxidative phosphorylation, the level of glycolysis, the metabolism of arginine, which is characterized by the formation of nitric oxide, and the oxidation of fatty acids are reduced.

## Data Availability

Not applicable.

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
