# Peer review of "The Influence of Mitochondrial-DNA-Driven Inflammation Pathways on Macrophage Polarization: A New Perspective for Targeted Immunometabolic Therapy in Cerebral Ischemia-Reperfusion Injury"

_ijms, 2021, doi:10.3390/ijms23010135_

Round 1

Reviewer 1 Report

The present work by Yu et al. elaborates the negative effect of stress-released mitochondrial DNA on the survival of neurons after ischemia damage. In this context, often described effects such as the activation of the immune system through the TLR system as well as the cGAS and STING signalling pathway are discussed. Unfortunately, this work lacks the debate about possible therapies introduced at the beginning. No new and experimental proposals were presented. 
Furthermore, the structuring of the individual chapters is not thought through and the titles are repetitive, so that the reader gets lost in perceived circles of discussion.
In addition to the chaotic structure, there is also a lack of care and the expected spell-checking. While some abbreviations are not explained at all, there are some that appear twice (e.g. STAT1 and STAT6). Sometimes authors are named with their first and last names, and a short time later this is discarded and only surnames are used again. 

Author Response

We gratefully thank the editor and all reviewers for their time spend making their constructive remarks and useful suggestions, which has significantly raised the quality of the manuscript and has enable us to improve the manuscript. Each suggested revision and comment , brought forward by the reviewers was accurately incorporated and considered .Below the comments of the reviewers are response point by point and the revisions are indicated.

Response to Reviewer 1 Comments

The present work by Yu et al. elaborates the negative effect of stress-released mitochondrial DNA on the survival of neurons after ischemia damage. In this context, often described effects such as the activation of the immune system through the TLR system as well as the cGAS and STING signalling pathway are discussed.

  1. Unfortunately, this work lacks the debate about possible therapies introduced at the beginning.

Thanks to the reviewer for such useful suggestion. We will add the following in the introduction

After reperfusion, the activation and infiltration of inflammatory cells, and the synthesis and secretion of adhesion molecules form a mutually promoting cascade reaction. In the primary and secondary injury stages of stroke, inflammation plays a major role in injury. At this stage, the curative effect is mainly achieved by blocking the inflammatory cascade, including therapeutic hypothermia, hydrogen sulfide treatment and the use of cyclosporine, cannabinoids, superoxide dismutase, metformin, and stem cell treatments.line58-64

  1. No new and experimental proposals were presented. 

Thanks to the reviewer for such useful suggestion. We have supplemented the experimental methods used in Table 1.

Table 1. mt-DNA driven inflammation & macrophage metabolism targeted therapy for Cerebral ischemia reperfusion injury

Target

Inhibitor

Mechanism

models

Reference

TLRs

Garcinol

TLR4-NFκB↓

middle cerebral artery occlusion/reperfusion (MCAO/R)、oxygen glucose deprivation and reperfusion (OGD/R)

[113]

Kudiezi injection

TLR4-NFκB↓

rat models of transient middle cerebral artery occlusion (tMCAO)

[114]

Pregabalin

HMGB1/TLR4--NFκB↓

Middle cerebral artery occlusion (MCAO) model

[115]

propofol

TLR4-NFκB↓

retinal ischemia reperfusion injury (RIRI)

[116]

Lactoferrin

TLR4 related pathways↓

anoxia and reoxygenation cell model、Institute for Cancer Research (ICR) mice

[117]

Lixisenatide

TLR4-NFκB↓

P38/ERK↑

NA

[118]

Tangeretin

Inflammatory cytokines  brain injury markers↓

mice model of cerebral ischemia/reperfusion injury

[120]

QiShenYiQi

TLR4-NFκB↓

OGD/R

[121]

nobiletin

Akt/mTOR↑

TLR4-NFκB↓

MCAO

[122]

Argon

TLR2/TLR4/STAT3/NF-κB ↓

retinal ischemia reperfusion injury (IRI) in rats

[124]

CX-10

Nrf2/AE

TLR/NF-κB

rat models of middle cerebral artery occlusion/reperfusion (MCAO/R)

[125]

NLRP3

diazoxide

NLRP3 inflammasome activation ↓

transient middle cerebral artery occlusion (tMCAO) rat model、oxygen-glucose deprivation/reoxygenation (OGD/R)

[98]

Sulforaphane

NLRP3 inflammasome activation ↓

middle cerebral artery occlusion (MCAO) model

[126]

verapamil

NLRP3 inflammasome activation ↓

transient middle cerebral artery occlusion (MCAO)

[127]

IMM-H004

NLRP3 inflammasome activation ↓

pMCAO model of focal ischemia

[128]

hispidulin

AMPK/GSK3β/ NLRP3- pyrolysis↓

middle cerebral artery occlusion (MCAO)、oxygen-glucose deprivation/reoxygenation (OGD/R)

[129]

astilbin

AMPK/GSK3β/ NLRP3- pyrolysis↓

middle cerebral artery occlusion (tMCAO) model with C57BL/6 J mice、oxygen-glucose deprivation and reintroduction (OGD-R) model

[131]

adiponectin

peptide

AMPK/GSK3β/ NLRP3- pyrolysis↓

middle cerebral artery occlusion-reperfusion (MCAO/R) model in rats

[132]

SB216763

AMPK/GSK3β/ NLRP3- pyrolysis↓

middle cerebral artery occlusion-reperfusion (MCAO/R) model in rats 、oxygen-glucose deprivation/reoxygenation (OGD/R)

[133]

Spautin-1

Autophagy/NLRP3- pyrolysis↓

middle cerebral artery occlusion-reperfusion (MCAO/R) model in rats 、oxygen-glucose deprivation/reoxygenation (OGD/R)

[134]

l-Homocarnosine

NLRP3 expression ↓

Middle cerebral artery occlusion/reperfusion (MCAO/R) model in rats

[135]

PAP-1

M1 polarization↓

NLRP3 inflammasome activation ↓

Middle cerebral artery occlusion/reperfusion (MCAO/R) model in rats and oxygen-glucose deprivation/ reoxygenation (OGD/R) in primary microglia

[136]

TMEM59

NLRP3- pyrolysis↓

NLRP3-inflammation↓

middle cerebral artery occlusion (MCAO)、Oxygen-glucose deprivation/reperfusion (OGD/R)

[137]

MCC950

NLRP3- pyrolysis↓

oxygen-glucose deprivation/reoxygenation (OGD/R)

[140]

Chlorpromazine &promethazine 

NLRP3 inflammasome activation ↓

middle cerebral artery occlusion/reperfusion (MCAO/R) model in rats

[142]

Idebenone

NLRP3 inflammasome activation ↓

oxygen glucose deprivation and reperfusion (OGD/R)

[143]

resveratrol

NLRP3 inflammasome activation ↓

middle cerebral artery occlusion/reperfusion (MCAO/R) model in rats

[144]

CD21

HMGB1/TLR4--NFκB↓

CathepsinB/NLRP3 inflammasome

activation ↓

global cerebral ischemia-reperfusion model in mice

[145]

meisoindigo

TLR4--NFκB↓

Cathepsin B/NLRP3 inflammasome

Activation↓

M1 polarization↓

C57BL/6J mice middle cerebral artery occlusion (MCAO) stroke model、HT-22 and BV2 cells in vitro oxygen glucose deprivation/Reperfusion (OGD/R) model

[146]

JQ1

NLRP3- pyrolysis↓

NLRP3-inflammation↓

middle cerebral artery occlusion (MCAO) model

[147]

tomentosin

TLR&NLRP3 Inflammation-related pathways↓

rats Cerebral ischemia-reperfusion model、oxygen glucose deprivation/Reperfusion (OGD-R)

[148]

Anthocyanin

TLR&NLRP3 Inflammation-related pathways↓

ICR mice cerebral ischemic/reperfusion (I/R) injury

[149]

Procyanidins

TLR&NLRP3 Inflammation-related pathways↓

middle cerebral artery occlusion/reperfusion (MCAO/R)、Oxygen-glucose deprivation/reoxygenation BV2 cells

[150]

Salvianolic Acids

M2 polarization ↑

Neuronal apoptosis ↓

NLRP3- pyrolysis↓

middle cerebral artery occlusion/reperfusion (MCAO/R) model in rats、oxygen-glucose deprivation/reoxygenation (OGD/R) model

[151]

Buyang Huanwu Decoction

NLRP3- pyrolysis↓

rats focal cerebral ischemia and reperfusion model

[152]

Genistein (Gen)

NLRP3 inflammasome activation ↓

not applicable

[153]

cGAS/STING

25-HC

Autophagy ↑ STING degradation↑

liver I/R model

[157]

MicroRNA-24-3p

inhibit cGAS/STING

transient middle cerebral artery occlusion (tMCAO)

[158]

Liproxstatin-1

Inhibit cGAS/STING

not applicable

[155]

A151

cGAS expression ↓

not applicable

[159]

Macrophage metabolism

STF31

Glycolysis ↓

diabetic retinopathy (DR) model

[161]

2-DG

Glycolysis ↓

not applicable

[162]

HK1 inhibitor

Glycolysis ↓

middle cerebral artery occlusion/reperfusion (MCAO/R) model in rats

[163]

Astragaloside IV

PPARγ- M2 polarization ↑

rat traumatic brain injury (TBI) model

[164]

CsA

Mitophagy↓- Glycolysis ↓

M2 polarization ↑

not applicable

[11,12]

3-BU

Glycolysis ↓

OXPHOS ↓

not applicable

[165]

L-NMMA

Inhibit iNOS

not applicable

[166]

Midivi-1

Mitochondrial division↓- Glycolysis ↓

not applicable

[169]

  1. Furthermore, the structuring of the individual chapters is not thought through and the titles are repetitive, so that the reader gets lost in perceived circles of discussion.

Sorry to confuse the reviewer. We have revised some titles. In addition, we are willing to explain the structure of our article to the reviewers here.

1.The function of macrophages in cerebral ischemia-reperfusion

2.The inflammatory response is driven by mt-DNA in cerebral ischemia-reperfusion injury

3.The influence of mt-DNA-mediated inflammation on the metabolic pattern of macrophages

4.The therapeutic prospects of cerebral ischemia-reperfusion targeting mt-DNA-mediated inflammation cGAS/STING and microglia/macrophage metabolism

  1. In addition to the chaotic structure, there is also a lack of care and the expected spell-checking. While some abbreviations are not explained at all, there are some that appear twice (e.g. STAT1 and STAT6).

We apologize for such errors, and all typos are correctly presented in the revised manuscript. See line76 88 144 228 236 305.

  1. Sometimes authors are named with their first and last names, and a short time later this is discarded and only surnames are used again.

We apologize for such errors, and all typos are correctly presented in the revised manuscript. see line 120. line 124. line 328. line 452. line 465. line 473. line 480.line 513.

Reviewer 2 Report

Critique:  The influence of mitochondrial -DNA-driven inflammation 2

pathways on macrophage polarization: a new perspective for 3

targeted immunometabolic therapy in cerebral ischemia-reper- 4

fusion injury.

Sihand Yu, Jiaying Fu, Jian Wang, et al.

ijms-1499511

Overall Impression:  I have reviewed a number of manuscripts from Chinese institutions and the written English has always been a problem.  This manuscript is the best I have seen for the mastery of English.  There are a few minor corrections that need to be made in the English and will require some dedicated proof reading.  The review is comprehensive and the authors’ research work is woven into the various pathways involved in contributing to the reperfusion injurty. 

The major concern I have is that the conclusions and perspective section needs to include the authors’ opinion regarding what if any experimental treatments for controlling reperfusion injury are suitable for clinical trials.  What are the barriers that prevent clinical trials for these compounds at the present time?

Author Response

Response to Reviewer 2 Comments

Critique:  The influence of mitochondrial -DNA-driven inflammation 2

pathways on macrophage polarization: a new perspective for 3

targeted immunometabolic therapy in cerebral ischemia-reper- 4

fusion injury.

Sihand Yu, Jiaying Fu, Jian Wang, et al.

ijms-1499511

Overall Impression:  I have reviewed a number of manuscripts from Chinese institutions and the written English has always been a problem.  This manuscript is the best I have seen for the mastery of English.  There are a few minor corrections that need to be made in the English and will require some dedicated proof reading.  The review is comprehensive and the authors’ research work is woven into the various pathways involved in contributing to the reperfusion injurty. 

Response:It is an honor to receive such comments from reviewers. On behalf of all the authors, we would like to thank the reviewer for your efforts on this manuscript.

The major concern I have is that the conclusions and perspective section needs to include the authors’ opinion regarding what if any experimental treatments for controlling reperfusion injury are suitable for clinical trials.  What are the barriers that prevent clinical trials for these compounds at the present time?

Response:It is a great honor to receive such a pertinent comment from the reviewer.

In terms of experimental research, as we mentioned in the article,line 522-524 However some problems still exist, such as the heterogeneity of the polarization level of macrophages in the injured area, and the difficulty of defining the administration time in the acute and subacute phases in the experiment.

As there are more or less differences in clinical drug standards in various countries, we cannot give a very specific explanation very responsible. However, we have added the following content to the manuscript after discussion and literature review. line 526-line530

Although many experimental studies on ischemia-reperfusion injury have been carried out around the world, more molecular mechanisms related to injury have not been explored in depth; And there is a lack of effective drugs and therapies for precise treatment of it. moreover the blood-brain barrier will reduce the efficacy of drugs to a certain extent.

Round 2

Reviewer 1 Report

The authors have contributed important extensions to this work. I have no further remarks to contribute.